# Technological Quality, Amino Acid and Fatty Acid Profile of Broiler Meat Enhanced by Dietary Inclusion of Black Soldier Fly Larvae

**DOI:** 10.3390/foods10020297

**Published:** 2021-02-02

**Authors:** Jessica de Souza Vilela, Tharcilla I. R. C. Alvarenga, Nigel R. Andrew, Malcolm McPhee, Manisha Kolakshyapati, David L. Hopkins, Isabelle Ruhnke

**Affiliations:** 1School of Environmental and Rural Science, Faculty of Science, Agriculture, Business and Law, University of New England, Armidale, NSW 2351, Australia; mkolaksh@myune.edu.au; 2Livestock Industries Centre, NSW Department of Primary Industries, Armidale, NSW 2351, Australia; tharcilla.alvarenga@dpi.nsw.gov.au (T.I.R.C.A.); malcolm.mcphee@dpi.nsw.gov.au (M.M.); 3Insect Ecology Lab, Zoology, University of New England, Armidale, NSW 2351, Australia; nandrew@une.edu.au; 4Centre for Red Meat and Sheep Development, NSW Department of Primary Industries, Cowra, NSW 2794, Australia; david.hopkins@dpi.nsw.gov.au

**Keywords:** eicosapentaenoic acid, insects, lauric acid, meat quality, poultry, omega-3, PUFA, sustainability

## Abstract

We evaluated the effects of full-fat black soldier fly larvae (BSFL) on broiler carcass composition, cut yield, and breast meat quality. Broilers were fed for 42 days with up to 20% dietary inclusion of BSFL (0, 5, 10, 15, and 20%). On day 42, 120 broilers were slaughtered, and images were taken using computed tomography. Breasts, drumsticks, and thighs were collected for cut yield determination. The pH, color, lipid oxidation, cooking loss, shear force, amino acid profile, and fatty acid profile of the breast meat were assessed. There was no dietary effect on carcass composition or meat quality parameters except for fatty and amino acids compositions. When 20% BSFL was included in the diet, individual fatty and amino acids, such as lauric and myristic acids, aspartic acid, glutamine, and lysine, increased by 22.0-, 5.50-, 1.08-, 1.06-, and 1.06-fold, respectively (*p* < 0.05). Although total polyunsaturated fatty acids decreased, eicosapentaenoic fatty acids (EPA) increased by 78% in the 20% BSFL inclusion group. In conclusion, up to 20%, dietary full-fat BSFL did not affect key meat characteristics but positively increased the levels of the health-claimable omega-3 fatty acid EPA.

## 1. Introduction

The quality of broiler meat has substantive economic impact as poultry meat accounts for the highest quantity of consumed meat globally—an average of 30.6 kg per person per year [1]. Consumer preference for chicken meat is driven not only by its relatively low purchase price and a lack of religious barriers, but also because it is a leaner source of animal protein compared to pork, beef, and lamb meat [2,3]. This preference for leaner meat sources reflects the awareness and demand for healthy food. For example, skinless chicken breast meat has lower levels of saturated fatty acids and total fat: 0.3–1.2 g per 100 g of meat, compared to raw beef steak cuts (1.8–4.3 g per 100 g of meat), raw beef loin (1.4–3.3 g per 100 g of meat), and raw pork loin (1.6–4.7 g per 100 g of meat) [4]. 

In animal nutrition, full-fat or defatted black soldier fly larvae (BSFL) have been used to partially replace common dietary ingredients such as soybean meal and soybean oil in poultry diets [5,6,7]. BSFL and soybean meal have high crude protein content, ranging from 25 to 53% in full-fat BSFL and from 45.1 to 49.5% in soybean meal [8,9]. 

The fatty acid profile of soybean oil differs from BSFL and other insect fat. Soybean oil has a lower quantity of saturated fatty acids (16.2%) and a higher quantity of polyunsaturated fatty acids (57.5%) compared to the fat from insects such as that from BSFL, which has been reported to contain 75% of saturated fatty acids and 12.9% of total polyunsaturated fatty acids [7,10]. 

Black soldier fly larvae are commonly used due to their ability for mass production (large-scale and continuous production of standardized products), the favorable production requirements (low use of space, water, energy resources, rearing substrate requirements), and the desirable nutritional composition, which includes not only favorable amino acid levels, but also antimicrobial properties [11,12,13,14,15,16].

While up to 20% BSFL can be fed to poultry without compromising the bird’s health or performance, the impact on poultry meat parameters relevant for human health is yet to be determined [5,11,14,17]. For example, BSFL can be composed of up to 57.8% fat, which may modify the intramuscular fat and fatty acid profile of broiler meat, thus influencing the diet of humans [8,17,18]. BSFL fat contains predominantly saturated fatty acids, particularly lauric acid with 12 carbon molecules and 0 double bonds (C12:0), representing up to 52% of the total fatty acids [8,19]. Lauric acid has demonstrated some anti-inflammatory and other positive effects modulating breast and endometrial cancer cells, as well as the human gut microbiota [20,21]. The dietary fatty acid composition significantly modulates the fat quality in animal’s tissues. For example, recent studies have demonstrated that the inclusion of partially defatted BSFL or BSFL fat in poultry, rabbit, and aquatic animal diets enhanced the fatty acid profile of the particular meat products, increasing total saturated fatty acids and lauric acid in broiler breast meat, fish fillet, and rabbit leg meat [6,22,23]. However, the impact of the maximum inclusion level (representing up to 20% of full-fat BSFL in a balanced diet for broiler diets) on broiler carcass composition, cut yield, and breast meat quality has not been established to date. It is relevant to establish this impact as the fatty acid composition of the broiler feed can affect sensory meat quality-related parameters such as lipid oxidation, pH, meat color, and the shelf life of broiler’s breast meat as a high percentage of unsaturated fatty acids in meat products is associated with higher oxidation levels and subsequently rancidity [24,25]. 

Therefore, this research aimed to examine the effects of various inclusion levels of full-fat BSFL in broiler diets on carcass and breast meat quality, focusing particularly on the fatty acid composition and shelf life parameters. We hypothesized that the BSFL dietary inclusion from day 2 to day 42 of broilers’ life would affect their carcass composition and cut yield as well as the composition of the broiler breast meat through modifications of the fat and amino acid profiles, consequently affecting the health status of the meat and other meat quality parameters such as lipid oxidation, color, pH, cooking loss, and shear force. 

Our specific questions concern, first, whether the dietary inclusion of BSFL would affect the carcass composition or cut yield by increasing the muscle, fat, or bone proportion in the carcasses or whether it would increase the proportional size of the cuts (breast, drumstick, and thigh); second, whether the BSFL dietary inclusions would affect breast meat quality parameters such as pH, meat color, lipid oxidation, cooking loss, or shear force; and third, whether the BSFL composition would affect fatty and amino acid compositions and the health status of the breast meat when included at up to 20% in broiler diets.

## 2. Materials and Methods 

This research was conducted under animal ethics approval granted by the University of New England’s Animal Ethics Committee (approval no. 18–084). 

### 2.1. Animal Experiment

Four hundred sexed male chicks (Ross 308) were obtained at one day of age, and ten chicks were allocated per cage, allowing for 0.044 m^2^ per bird. The average initial body weight of the broilers was determined and used to randomly allocate the chicks into groups with comparable total body weight. On day two, the 40 cages were randomly allocated to one of five dietary treatment groups, resulting in eight replicates per treatment. All diets were formulated targeting comparable energy levels and digestible amino acid values for each of the three feeding phases (starter, grower, and finisher diet) meeting or exceeding Ross 308 (2016) nutrient requirements. The starter diet was offered from day 2 to day 10 of broilers’ age for Treatments (T) 1, 2, 3, 4, and 5, which included 0, 2.5, 5, 7.5, and 10% of BSFL in the diet, respectively. The grower and finisher diets were fed from 11 to 21 and 22 to 42 days of age when T 1, 2, 3, 4, and 5 included 0, 5, 10, 15, and 20% of BSFL, respectively. BSFL (Karma^3^ Ptd Ltd.; Melbourne, Victoria, Australia) were used in their full-fat form and previously dried at 80 °C by the provider. While BSFL contained 40% crude protein and 32.5% crude fat, their use partially replaced various protein sources (soybean meal, meat-and-bone meal) and vegetable fats (canola and cottonseed oil), resulting in the comparable energy levels of the experimental diets. The fatty acid profiles of BSFL and the diets are outlined in Table 1. Further details of the BSFL chemical composition, amino acid profile, and fatty acid profile, and the quantitative component composition and calculated nutrient composition of the experimental mixtures of the diets at different growth phases are described in de Souza et al. [5]. All experimental diets and drinking water were provided ad libitum using trough feeders and nipple drinkers.

On day 42 of the experiment, three broilers per cage (24 broilers/treatment) were randomly selected, individually weighed, stunned, and humanly slaughtered using cervical dislocation. The broiler carcasses were eviscerated by removing the gastrointestinal tract and organs (liver, spleen, heart, bursa) and fat pad. The remaining body components (carcass with head, neck, feet, and feathers) were individually tagged and scanned using computed tomography (CT) for carcass tissue composition determination. 

### 2.2. Carcass Composition and Cut Yield

Carcass images were obtained using a CT scanner (GE Healthcare, HiSpeed Qx/I, Milwaukee, WI, USA). The eviscerated chickens were placed on the CT scanner bed in a dorsoventral position. An average of 32 scans of the carcasses (with head, neck, feet, and feathers) were taken per animal. The radiation tube operated at 120 kV (tube voltage) and 100 mAs (radiation intensity), using 5 mm thickness, 10 mm spacing, a pitch of 1.5, and a field of view of 500 mm. The directionality of the scans was craniocaudal. Images were processed using an open-source image processing package (ImageJ bundled with 64-bit Java 1.8.0_112 downloaded at https://imagej.net/Downloads; Figure 1). The estimates of lean and fat tissue weights (g) from scanned images were then calculated using in-house software [26].

### 2.3. Meat Quality Analysis 

#### 2.3.1. Color and pH Measurement 

The pH of the samples was measured in duplicates using a hand-held pH meter (IJ44C probe, Ionode Pty Ltd., Tennyson, QLD, Australia) with an integrated temperature sensor (WP-80 Waterproof pH-mV-Temperature Meter, TPS, Brendale, NSW, Australia). The pH meter was calibrated at room temperature using buffers at pH 4.0 and 6.88. Color measurements were taken in parallel alignment to the fiber position, in triplicates, and on a freshly cut surface after 30–40 min of blooming using a Minolta Chroma Meter CR-300 (Minolta Co., Ltd., Osaka, Japan). The instrument was calibrated using a white tile (Y = 93.3, x = 0.3135, y = 0.3198; Minolta Co., Ltd., Osaka, Japan) using illuminant D-65, with a 2° standard observer. A total of three readings were taken on each steak and averaged. Values of lightness (*L**), redness (*a**), and yellowness (*b**) were recorded. Chroma (*C**) and hue angle (*h**) were calculated as a function of *a** and *b** according to [30], following the formulas
$$C*=\sqrt{\left(a{*}^{2}+b{*}^{2}\right)}\;h*=\tan^{-1}\left(\frac{b*}{a*}\right).$$

#### 2.3.2. Cooking Loss and Shear Force 

These methodologies were based on Downing et al. [31]. Meat blocks of approximately 65 g were weighed (fresh sample weight), vacuum packed, and stored at −20 °C for subsequent cooking loss and shear force analysis. For cooking loss, the frozen samples were placed into a water bath (Model: BTC 9090, Thermoline, Sydney, NSW, Australia) set at 85 °C for 25 min and then placed under cold running tap water 30 min to stop the cooking process. Samples were then removed from the bag, wrapped in paper towels to remove the excess of water, and weighed (cooked sample weight) for cooking loss determination. Meat samples from cooking loss determination were stored in individual zip lock bags at 4 °C overnight for subsequent shear force analysis. 

Shear force analysis was performed using a Lloyd Instruments LRX Materials Testing Machine fitted with a 500 N load cell (Lloyd Instruments Ltd., Hampshire, UK). Briefly, five to seven subsamples with a rectangular cross-section of 15 mm width and 6.66 mm depth (1 cm^2^) were cut from each block, with fiber orientation parallel to the long axis right angles to the shearing surface. The force required to shear through the clamped subsample with a triangulated 0.64 mm thick blade pulled upward at a speed of 100 mm/min at a 90 degree angle to the fiber direction and was expressed as kg peak force. Values of the kg peak force were recorded, and the mean value obtained from the subsamples was converted to N for statistical analysis. The conversion of kg values into Newton values multiplied 1 kg to 9.81 as 1 kg equals 9.81 N.

#### 2.3.3. Lipid Oxidation 

The methodology used in this analysis is fully described by Holman et al. [32]. In brief, a portion of approximately 5 g of breast meat obtained from 80 broilers (2 broilers per cage, 16 broilers per treatment) was vacuum packed and stored at −80 °C to prevent oxidation. Thiobarbituric acid-reactive substances (TBARS) were used to determine the lipid oxidation of the broiler breasts. Approximately 100 mg of frozen sample was added to 500.0 µL radio-immunoprecipitation assay (RIPA) buffer (Item No. 10010263, Cayman Chemical Company Ltd., Ann Arbor, MO, USA) and homogenized using micro-tube pestles [30]. The supernatant was isolated, and TBARS contents were determined as per the TBARS (TCA Method) Assay Kit colorimetric protocol (Item No. 700870, Cayman Chemical Company Ltd., Ann Arbor, MO, USA). A benchtop spectrometer (model FLUOstar OPTIMA™, BMG Labtechnologies, Melbourne, VIC, Australia) was set to measure absorbance at 540 nm. Technical duplicates were averaged, and data were expressed as mg malondialdehyde (MDA) per kg of wet sample.

#### 2.3.4. Amino Acid and Fatty Acid Analysis 

Samples of breast meat (~45 g) from 80 birds (two broilers per cage, 16 broilers per treatment) were cut into pieces and stored at −20 °C. The samples were freeze-dried (FD–PILOT7–12 Series, Dynavac, Sydney, Australia) at −55 °C for seven days and then weighed again for dry matter determination. Freeze-dried samples were ground using a multigrinder (Sunbeam Multigrinder II, Botany, NSW, Australia), well homogenized, and divided into two containers for amino acid and fatty acid profile determination. 

Analysis of amino acid profile was performed using liquid hydrolysis in 6 M hydrochloric acid to extract the amino acids from the protein, followed by the High-performance Liquid Chromatograph (HPLC) procedure [33]. Briefly, samples underwent 24 h of liquid hydrolysis in 6 M hydrochloric acid at 110 °C. Under these conditions, asparagine was hydrolyzed to aspartic acid and glutamine to glutamic acid; therefore, the reported amount of these acids is the sum of those respective components. After hydrolysis, all amino acids were labeled using the Waters AccQTag Ultra chemistry (Waters Corporation, Totowa, NJ, USA), following the supplier’s recommendations, and analyzed using Waters Acquity Ultra Performance Liquid Chromatography (UPLC; Waters Corporation, APC™, Milford, MA, USA). 

Total lipids of BSFL, as well as the fatty acid profile of the experimental diets (Table 1) and breast meat samples, were determined according to a modification of the method described by Folch et al. [34] and the fatty acid profile was analyzed as per a modification of the method described by Clayton et al. [35]. In summary, ~10 mg of lyophilized samples (BSFL, feed, and breast meat) were impregnated with methanol. Glass culture 8 mL tubes were used to mix the lyophilized samples with 2.0 mL of methanol/toluene (4:1 *v*/*v* that contained 10.0 μg/mL each of C13:0 and C19:0 internal standards). Atherogenic and thrombotic indexes were calculated according to Ulbricht and Southgate [36]. 

The methylation of fatty acids was conducted by the addition of 200 μL of acetyl chloride followed by 60 min of incubation in a heating block (at 100 °C). After cooling, a 6% potassium carbonate solution (5 mL) was individually added to all samples to impede addition reactions. The upper toluene supernatant layer was separated after centrifuging the solution, then 2 mL was transferred into a glass gas chromatography (GC) vial with a Teflon-lined screw-cap until the next step of the analysis. Then, individual fatty acid methyl esters were identified using an Agilent 6890 N GC with a flame ionization detector (FID; Model 6890 N, Agilent, Santa Clara, CA, USA). To carry the gas in the GC analysis, helium was used (total flow rate of 12.4 mL per min, a split ratio of 10:1, and a column flow of 0.9 mL per min) in a focus inlet liner (4 mm i.d., no. 092002, SGE Analytical Science, Victoria, AUS), and the inlet injection volume, pressure, and temperature were, respectively, 2.5 μL, 107.8 kPa, and 250 °C. The oven temperature was initially set to 150 °C, held for 30 s, and then gradually increased at 10 °C per minute to 180 °C, 1.5 °C per min to 220 °C, and then finally to 260 °C and held for 5 min. The total oven run time per sample was 36.5 min. The flame ionization detector temperature, gas flow rates of H, instrument air, and N_2_ make-up gas were 280 °C, 35 mL per min, 350 mL, and 30 mL per min, respectively. Commercial standards (Nu-Chek Prep, Minnesota, USA; Sigma-Alrich, Missouri, USA; Supelco, Pennsylvania, USA) and published data [37] were used to recognize the fatty acid methyl ester peaks. A three-point standard curve was performed using Agilent Chemstation C.01 and Microsoft Excel was used to calculate fatty acid methyl ester concentrations.

### 2.4. Statistical Analysis

A linear mixed model was performed in RStudio (© 2009–2020 RStudio, PBC, Version 1.3.1093) to analyze the effects of BSFL levels (0, 5, 10, 15, and 20%) with BSFL levels as fixed effects, BW as a covariate, and cage as a random term for muscle, total fat, total bone, breast, drumstick, and thigh. Analysis of the ratios (muscle/body weight, fat/body weight, breast/body weight, drumstick/body weight, and thigh/body weight), meat quality parameters, and amino acid and fatty acid profiles did not have body weight as a covariate. Within the meat quality traits, the analysis of breast meat color included pH as a covariate and cage as a random factor. All two-way interactions were examined and were non-significant (*p >* 0.05). A plot of the residuals indicated that total fat required a square root transformation, and the ratios each required to be square transformed, where back-transformed values are reported. 

## 3. Results

### 3.1. Carcass Composition and Cut Yield

No effects of the dietary BSFL inclusions were found in the muscle, total fat, total bone, muscle/body weight, and fat/body weight ratios. It was observed that there was a significant (*p* = 0.046) increase in total bone between the control group (T1) and the group fed 10% BSFL (T3), but this was not observed for treatments T2, T4, and T5 (Table 2 and Table 3). 

There were no effects of the BSFL dietary inclusions on the broiler breast meat, drumstick, thigh, or any cut yield evaluated such as breast/body weight, drumstick/body weight, and thigh/body weight ratios (Table 4). 

### 3.2. Meat Quality

#### 3.2.1. pH, Color, Lipid Oxidation, Cooking Loss, and Shear Force

There were no effects of the dietary level of BSFL on meat pH, color parameters (*L**, *a**, *b**, *C**, and *h**), or lipid oxidation of breast meat (Table 5). There were also no effects of the dietary level of BSFL on the cooking loss and shear force of breast meat (Table 5). 

#### 3.2.2. Amino Acid Profile

The increasing levels of BSFL in the diet decreased the amino acid serine in the 10% BSFL inclusion broiler group (Table 6). However, the inclusion of 15% BSFL in the diet significantly increased aspartic acid, glutamine, and lysine in broiler breast meat (Table 6).

#### 3.2.3. Fatty Acid Profile

##### Saturated Fatty Acids 

An increase in the dietary level of BSFL resulted in increases in decanoic acid (C10:0), lauric acid (C12:0), myristic acid (C14:0), pentadecanoic acid (C15:0), and isoheptadecanoic acid (C17:0) in the breast meat samples (Table 7). The maximum increase in decanoic acid (C10:0) was from 5.78 to 9.91 mg/100 g, while lauric acid (C12:0) increased up to 22-fold, from 12.2 to 268.8 mg/100 g. Myristic acid (C14:0) increased up to 5.5-fold, from 18.7 to 103.0 mg/100 g. Isoheptadecanoic acid (C17:0iso) also increased from 0.39 to 1.75 mg/100 g in broiler breast meat with the 20% BSFL dietary inclusion. 

An increase in the dietary level of BSFL resulted in a linear decrease in stearic acid (C18:0), eicosanoic acid (C20:0), and behenic acid (C22:0) of broiler breast meat when broilers were fed the maximum inclusion of BSFL (20%) in the diet (Table 7). 

##### Monounsaturated Fatty Acids 

The inclusion of BSFL in the diet did not affect monounsaturated fatty acids (MUFAs). An increase in the dietary level of BSFL resulted in increases in trans-palmitoleic acid (C16:1ω-7t), palmitoleic acid (C16:1ω–7), myristoleic acid (c14:1ω-5), and nervonic acid (C24:1ω–9) when broilers were fed with up to 20% of BSFL in the diet (Table 7). 

An increase in the dietary level of BSFL resulted in a decrease in C18:1ω-9trans (elaidic acid). C18:1ω-7trans (trans-vaccenic acid) also presented a significant reduction with the BSLF inclusion (Table 7). 

##### Polyunsaturated Fatty Acids 

An increase in the dietary level of BSFL increased the level of eicosapentaenoic acid (EPA; C20:5ω–3) by 78.6% at the maximum of 20% BSFL. Mead acid (C20:3ω–9) also increased with BSFL and increased by 61.5% at the maximum of the 20% BSFL inclusion compared to the control diet. The increase in the dietary level of BSFL resulted in a decrease in adrenic acid (C22:4ω–6), eicosadienoic acid (C20:2ω–6), docosapentaenoic acid (C22:5ω–6), and hexadecadienoic acid (C18:3ω–4), (C16:2ω–4), as well as in the total quantity of polyunsaturated fatty acids (PUFAs). The total quantity of PUFAs significantly decreased by 37.4% with increasing use of dietary BSFL. The essential fatty acids linoleic acid (C18:2ω-6) and alpha-linolenic acids (ALA; C18:3ω–3), as well as docosahexaenoic acid (DHA; C22:6ω-3), were not significantly affected by the BSFL dietary inclusions (Table 7). 

Increased dietary levels of BSFL did not affect the sum of ω-3 (Σ PUFA ω-3) fatty acids. The ω-6 (Σ PUFA ω-6; Table 7) fatty acids decreased with the BSFL inclusions. The significant ω-6 (Σ PUFA ω-6) reduction was from 674 to 407 mg/100 g. The ω-6/ω-3 ratio also reduced when the dietary inclusion of BSFL increased. The total increase in the ω-6/ω-3 ratio decreased from 17.6 (treatment 1) to 11.2 (treatment 5) (Table 7). 

##### Trans-Saturated Fatty Acids, Atherogenicity and Thrombogenicity Indexes 

The sum of trans-fatty acids reduced from 8.27 mg/100 g in treatment 1 to 5.77 mg/100 g in treatment 3 with the BSFL dietary inclusions. Atherogenic and thrombotic indexes increased with the BSFL dietary inclusions. The atherogenic index increased from 0.46 to 1.13, while the thrombotic index increased from 0.03 to 0.17 in treatments 1 and 5, respectively. 

## 4. Discussion

### 4.1. Carcass Composition, Cut Yield, Meat Quality Parameters, and Amino Acid Profile 

Neither carcass tissue composition, cut yield (breast, drumstick, and thigh) ratios, nor the majority of meat quality parameters such as breast meat color, pH, lipid oxidation, cooking loss, or shear force were affected by the BSFL inclusion levels. Besides the yellowness index variation, there was no statistical difference between treatments. The increase in yellowness is probably not surprising given the increase in fat levels and potentially an increase in specific pigments, but the effect was not judged likely to lead to a detrimental impact on consumer acceptability. Other studies [7] have found no effect of the BSFL fat inclusion on meat pH or color parameters when feeding up to 6.9% BSFL fat to broilers for 35 days but bear in mind the much lower level of BSFL inclusion in the diet. Further, equivocal effects of the BSFL inclusion on meat quality were reported by other authors. For instance, Popova et al. [38] included 5% during the finishing period of broilers (14 to 35 days) of partially defatted and full-fat BSFL. They reported a decrease in pH and a lighter color of broiler breast meat from broilers fed BSFL. Cullere et al. [39], when investigating quail meat quality, did observe an effect on muscle pH when 10 or 15% of defatted BSFL (containing 14.8% fat) with total dietary fat of 5.2 or 4.6% were fed, resulting in a significant reduction in pH (from 5.76 to 5.67%), accompanied by a significant increase in cooking loss from 24.1 to 28.1%. These discrepancies within study results may be related to differences in the type of BSFL used (e.g., BSFL fat, defatted BSFL, or full-fat BSFL). The BSFL inclusion levels also might have influenced the outcomes. Furthermore, differences in the diet composition between studies could also be a factor. The discrepancies may have also happened due to the metabolic differences within species of the commercial broilers used in some studies and the quails used in the other studies. 

Parameters of cooking loss, shear force, meat color, and pH values ranged within the physiologic values expected for chicken breast meat, with some exceptions. For instance, values of cooking loss in this study were roughly around 30%, while in other studies, these values were reported around 18% [32,40]. Our shear force values ranged from 21 and 23 N compared to the ranges of 11 to 12 N and 15 to 16 N in other studies [31,40]. These differences in shear force and cooking loss could be due to numerous reasons, such as differences in broiler age, breed, and/or environmental conditions applied to the broilers in the different experiments. Values of color were similar to the ones previously reported by Downing et al. [31] and Elshafaei et al. [40]. 

In this study, the amino acid profile of the experimental diets was comparable between treatment groups. However, there was a maximum 1.06-fold increase (from 68.6 to 72. 5 mg/kg) in lysine between the meat obtained from the control (T1) and the 15% BSFL inclusion groups (T3). There were also higher non-essential amino acid levels such as aspartic acid and glutaminic acid in the breast meat obtained from the 5%, 10%, and 15% BSFL inclusion groups. Serine was significantly lower in the meat obtained from broilers of the 10% BSFL group compared to the others. Cullere et al. [39] included 10 and 15% of defatted BSFL in quail diets and reported similar results such as increases in threonine, alanine, aspartic acid, glutamic acid, serine, and tyrosine concentrations in their meat. There is a lack of studies related to the influence of BSFL inclusions in broiler diets on the amino acid profile of broiler meat. 

### 4.2. Fatty Acid Profile 

The dietary fatty acid profile was highly modified by the BSFL inclusions; out of the 47 fatty acids identified in the breast meat, 23 were significantly modified by the BSFL inclusions in the diets. Diets containing BSFL tended to have a fatty acid profile more similar to the fatty acid profile of BSFL fat. As demonstrated in this research and by other studies, the main component of BSFL fat is lauric acid, and it has strong antimicrobial properties, which help the BSFL to cope with potential environmental threats such as pathogenic microorganisms present in organic waste streams [19,41]. While lauric acid has been considered unfavorable for human consumption due to its increase in low-density lipoprotein (LDL) cholesterol and, subsequently, cardiovascular diseases, it also has a beneficial regulatory function against breast, colon, and endometrial cancer cells [20,42]. Similarly, myristic acid is also prominent in BSFL fat composition and is linked to cardiovascular disease [43]. The maximum dietary inclusion of 20% full-fat BSFL (32.5 ± 3.5% fat) with a total BSFL fat inclusion of 6.4% in the experimental broiler diets caused increases of up to 4.2-fold (from 15.7 to 65.3 mg/100 g) in the total saturated fatty acids, 226-fold (from 0.13 to 29.4 mg/100 g) in the lauric acid, and 13.4-fold (from 0.37 to 4.97 mg/100 g) in the myristic acid dietary concentrations. It is well known that the dietary fatty acid profile modulates the fatty acid profile of the meat. Thus, a series of significant changes affected breast meat fatty acid composition by feeding the BSFL diets: 

Significant increases in dietary lauric and myristic acid (C12:0; C14:0) concentrations, which exhibited 22- and 5.5-fold increases, respectively, increased the total saturated fatty acid (Σ SFA) concentration in broiler breast meat when up to 20% BSFL inclusion was included in the diets. Schiavone et al. [44] reported similar fatty acid increases when 6.9% BSFL fat was added in broiler diets from days 1 to 35 of broilers’ age, observing increases of 94.4-fold (from 0.09 to 8.5%) in lauric acid and 41% in myristic acid, and also an increase of 35% in the total saturated fatty acids (from 32.2 to 43.5%) in the concentration of the total fatty acids identified in the breast meat samples. The increase in individual saturated fatty acids in the fat composition of the breast meat caused a reduction in the breast meat concentration of the monounsaturated fatty acid oleic acid (18:1ω-9) and the polyunsaturated essential fatty acid linoleic acid (18:2 ω-6), as well as total PUFAs fat, including total ω-6 fatty acids. The concentration of trans-vaccenic acid (C18:1ω-7trans) also decreased with the BSFL dietary inclusions. This fatty acid is known for its positive effects on human health by reducing cancer cells and supporting people with diabetes type 2 by increasing insulin secretion [45,46]. The European Food Safety Authority reported that the recommended intake for saturated fatty acids should be as low as possible as the intake of saturated fatty acids is positively related to the blood levels of LDL, cholesterol, and cardiovascular disease occurrence [47]. However, some studies have reported no relationship between saturated fatty acid intake and cardiovascular disease occurrence [48,49]. 

In contrast, PUFAs and especially the ω-3 long-chain fatty acids EPA (20:5 ω-3) and DHA (22:6 ω-3) have been related to numerous health benefits such as a reduction in cancer risks, inflammation, and cardio protective activities, as well as improvements in brain function [50,51]. Desirable levels of ω-3 fatty acids in the human diet are still not well established and they have been reported to depend on individual characteristics such as gender, age, and physiological status [52,53]. 

It is likely that most people do not consume a sufficient amount of ω-3 for the greatest health benefits. It has been reported that global recommendations of total ω-3 consumption for children and pregnant women for health benefits, neurodevelopment, and birth weight are between 200 and 300 mg and up to 2.7 g/per day [52,53]. The total ω-3 quantity of broiler breast meat did not differ between treatments, ranging from 36.7 in the 20% BSFL-fed group to 37.8 mg/100 g in the control breast meat. This indicates that 100 g of broiler breast meat fed on BSFL would supply the daily recommended consumption of total ω-3 fatty acids for humans. Furthermore, products rich in ω-3 fatty acids can impact the consumer’s decision when considering its purchase [54,55], and various specialty products such as ω-3-enriched eggs or ω-3-enriched dairy products have been marketed for decades. In this study, the DHA (22:6ω-3) concentrations in the broiler breast meat were not affected by the BSFL dietary inclusions as the diets had a similar or the same DHA concentration. In contrast, the EPA (20:5ω-3) dietary concentrations increased by 4-fold when up to 20% BSFL inclusion was included in the diets, consequently also increasing the EPA concentration in the breast meat of these broiler groups fed 20% BSFL. The maximum EPA increase in the broiler breast meat was a 79% increase (from 1.38 to 2.51 mg EPA/ 100 g breast meat) when fed a diet containing 20% BSFL. However, the sum of DHA and EPA in the broiler breast meat was much lower (maximum of 8.22 mg/per 100 g of broiler breast meat in the 20% BSFL group) compared to values observed in lamb meat (140 mg/per 100 g of lamb meat) when lamb feed was supplemented with 1.92% of a commercial alga (DHA-Gold^TM^), reported by Hopkins et al. [56], and the sum of total saturated fatty acids increased by over 30% in the broiler breast meat from broilers fed 20% BSFL. 

The ω-6 arachidonic acid (AA; C20:4ω-6) may increase inflammation processes, and it has been reported to be significantly reduced in breast meat even with equivocal dietary concentrations of this fatty acid [56]. The AA concentration was very similar in all diets (varied from 0.02 to 0.04 mg of the diet), but the quantity of this ω-6 fatty acid significantly decreased its concentration in the breast meat from 59.2 in T1 compared to 45.6 mg/100 g chicken breast meat in T5. 

The quantity of desirable ω-3 long-chain fatty acids in broiler breast meat can affect lipid oxidation and shelf life. These values in the broiler breast meat are below the Australian dietary guideline recommendations. The Australian dietary guidelines recommend the minimum consumption of 250 mg of ω-3 long-chain fatty acids [57]. The sum of EPA and DHA in 100 g of the broiler meat, regardless of the BSFL inclusion, was still below the recommendations of daily consumption of these fatty acids. The presence of PUFAs and especially ω-3 fatty acids is desired for the health status of the meat, but it increases lipid oxidation in the meat product, thereby affecting shelf life [25]. With the levels of ω-3 fatty acids being relatively low and a 30% increase in total saturated fatty acids, it is not surprising that the lipid oxidation of the breast meat was unaffected. Lipid oxidation is one of the main meat quality parameters as an increase in lipid oxidation can affect the organoleptic characteristics of the meat [25,58,59]. In this study, increasing dietary BSFL inclusion levels and the subsequent decrease in the breast meat PUFAs altered neither lipid oxidation nor any other related meat quality parameters such as meat pH and color. 

The knowledge related to fatty acids’ role in human health is not limited to the concentration of fatty acids, but the ratio of the preferred ω-6/ω-3 fatty acids ratio remains unknown [60,61,62,63]. Nevertheless, many researchers have highlighted the importance of having the ω-6/ω-3 ratio below 5 for the health status of meat products [60,61,62,63]. In this study, the ω-6/ω-3 ratio of the broiler breast meat was reduced by the BSFL dietary inclusion from 17. 6 in the control broiler breast meat, which is currently considered high. This ratio decreased to 11.2 with the 20% BSFL inclusion treatment group, which is lower but still higher compared to the ideal value of >5. This decrease in the ω-6/ω-3 ratio in the breast meat of broilers fed the 20% BSFL diet is due to the reduction in total ω-6 fatty acids in the broiler breast meat caused by the BSFL inclusion in the diets but is a non-alteration in the total amount of ω-3 fatty acids. Trans-saturated fatty acids reduced in diets containing higher levels of BSL compared to diets containing 0 or 5% BSFL inclusion. 

The sum of trans-fatty acids varied from 8.3 in the control chicken breast meat to 5.8 mg/100 g in the 10% BSFL breast meat. This reduction in trans-fatty acids is positively related to the health status of the chicken breast meat as trans-fatty acids have been reported to be linked to coronary heart diseases [64]. However, there are no studies presenting results of the effects of insect or BSFL meal dietary inclusions on trans-fatty acids deposition in chicken breast meat to date. 

The atherogenic index reduced from 0.46 in control breast meat to 1.13 in the 20% BSFL inclusion breast meat. Further, the broiler breast meat thrombotic index increased when BSFL were included in the broiler diets. The increase was from 0.03 in control breast meat to 0.17 in the 20% BSFL inclusion breast meat. Attia et al. [65] reported similar values of the atherogenic index for frozen and fresh chicken meat. Thrombotic indexes of our study are lower compared to other studies (e.g., [65]). The lower thrombotic index values are possible due to the high levels of oleic acid and total ω-6 fatty acids of the experimental diets. 

The fact that most of the meat quality properties were not significantly affected by the 20% BSFL inclusion level suggests that BSFL can be included in broiler diets without compromising consumer acceptance. However, the impact of the modulated fatty acid composition in the broiler breast meat, such as the significant increase in saturated fatty acids and the accompanying decrease in PUFAs, has to be further studied to evaluate the impact of this fatty acid change on consumers’ health. While the impact of these fatty acids on the total intake of fatty acids in human diets may be negligible, the consequence for consumers’ perception needs further investigation. Another relevant aspect should be the consumers’ perception of chicken meat-fed insects. While a global trend of sustainable food sources can be observed, food neophobia may be related to insect use [66].

## 5. Future Perspectives 

Broiler breast meat parameters in all broilers fed up to 20% BSFL resulted in a meat quality suitable for human consumption. Enriching the BSFL rearing substrates with sources of ω-3 fatty acids, such as linseed or flaxseed oil, may modify the BSFL nutrient composition and therefore positively affect broiler breast meat composition. For instance, Oonincx et al. [67] decreased the ω-6/ω-3 ratio by adding 1% flaxseed oil in the rearing substrate of house crickets, lesser mealworms, and BSFL. The reductions found in the ω-6/ω-3 ratio when 1% of flaxseed oil was added in the rearing substrates were from 18:3 in the BSFL fat, from 36:7 in the house cricket fat, and 22:6 in the lesser mealworm fat. Further research is required to study the effect of full-fat BSFL on meat quality due to the storage time.

## 6. Conclusions

The effects of BSFL on meat quality parameters were mainly related to the changes in the fatty acid profile of the meat caused by the fatty acid profile of the BSFL. In this study, up to 20% BSFL inclusion did not affect broiler carcass composition or the cut yield ratio, or meat pH and color, cooking loss, shear force, or lipid oxidation of the broiler breast meat. Amino acids such as aspartic acid, glutamine, and lysine increased in the breast meat of broilers fed BSFL. However, there was a reduction in total PUFA’s, while an increase in EPA was observed, coupled with an increase in total saturated fatty acids and, in particular, lauric acid. The ω-6/ω-3 ratio of the broiler breast meat was decreased with the BSFL dietary inclusions. A 22-fold increase in lauric acid caused by high levels of this fatty acid in the BSFL might be beneficial to the health status of broiler breast meat due to the potential health benefits of lauric acid. The amount of total saturated fatty acids increased with the BSFL dietary inclusions by 30%, possibly beneficial to the shelf life of chicken breast meat, but unfavorable for human health. Based on the findings, it would appear that if up to 20% BSFL inclusion is added to a diet, then other components that will affect PUFAs should be considered.

## Figures and Tables

**Figure 1 foods-10-00297-f001:**
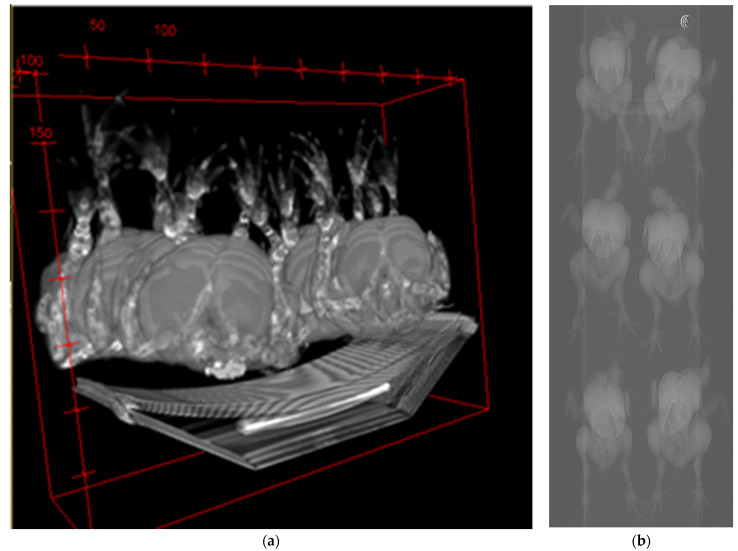
Computed tomography (CT) image of broiler carcasses for tissue composition determination: (**a**) three-dimensional view from the craniocaudal aspect and (**b**) dorsoventral view of six broiler carcasses arranged on the CT scanner bed. Immediately after scanning the broiler carcasses, breast muscles (pectoralis major and minor), drumsticks (muscle), and thighs (muscle) were removed from the carcasses and individually weighed to determine cut yield. Thigh, drumstick, and breast meat yields were expressed as a percentage of the live body weight as described previously [27,28,29]. The live body weight is described and published as individual body weight in de Souza et al. [5]. The paired breast muscles were labeled (treatment group, right and left side), placed pairwise in plastic bags, and stored overnight at 4 °C. On the following morning (approximately 24 h post-mortem), individual breast muscles were consistently partitioned for meat quality measurements (Figure 2).

**Figure 2 foods-10-00297-f002:**
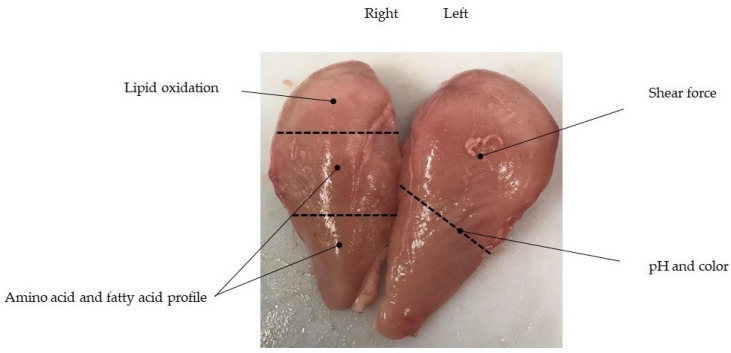
Sampling allocation of meat quality parameters of broiler’s breast muscle.

**Table 1 foods-10-00297-t001:** Fatty acid profile of experimental diets provided to the broilers at the starter, grower, and finisher dietary phases.

Fatty Acid (mg/100 g)	Starter Diets ^1^					Grower Diets ^2^					Finisher Diets ^3^				
	0%	2.50%	5%	7.50%	10%	0%	5%	10%	15%	20%	0%	5%	10%	15%	20%
*Saturated fatty acids (SFAs)*															
C8:0	0.10	0.09	0.09	0.04	0.06	0.07	0.09	0.07	0.08	0.09	0.09	0.09	0.11	0.1	0.09
C9:0	0.01	0.01	0.02	0.04	0.04	0.01	0.02	0.04	0.06	0.07	0.01	0.03	0.04	0.07	0.09
C10:0	0.11	0.25	0.49	0.79	0.86	0.11	0.38	0.85	0.94	1.28	0.10	0.63	0.64	1.27	1.60
C11:0	0.02	0.02	0.02	0.02	0.03	0.02	0.02	0.03	0.03	0.06	0.01	0.03	0.03	0.06	0.06
C12:0	0.08	2.51	7.03	12.8	14.0	0.09	5.54	14.3	17.8	22.7	0.13	9.55	10.6	23.5	29.4
C14:0	0.18	0.56	1.33	2.28	2.41	0.20	1.06	2.61	3.34	4.07	0.37	1.83	2.02	4.21	4.97
Iso–C15:0	0.03	0.03	0.03	0.02	0.03	0.02	0.02	0.03	0.04	0.05	0.02	0.03	0.03	0.05	0.05
Anteiso-C15:0	0.01	0.01	0.02	0.03	0.02	0.01	0.02	0.02	0.02	0.04	0.01	0.01	0.02	0.03	0.04
C15:0	0.07	0.08	0.09	0.09	0.09	0.08	0.08	0.10	0.12	0.11	0.08	0.08	0.09	0.12	0.12
C16:0	6.81	7.01	7.86	8.72	8.09	7.51	6.61	8.77	10.1	9.92	11.7	11.3	9.88	12.8	11.3
iso–C17:0	0.01	0.01	0.01	0.01	0.01	0.02	0.01	0.01	0.01	0.01	0.01	0.01	0.01	0.01	0.01
Anteiso-C17:0	0.02	0.03	0.04	0.05	0.04	0.03	0.03	0.04	0.04	0.06	0.02	0.03	0.03	0.05	0.05
C17:0	0.09	0.10	0.12	0.16	0.11	0.12	0.09	0.12	0.16	0.15	0.12	0.12	0.10	0.16	0.17
C18:0	1.82	1.82	1.90	2.13	1.65	2.30	1.54	2.00	2.22	2.09	2.26	2.02	1.61	2.30	2.20
C20: 0	0.25	0.22	0.18	0.16	0.11	0.34	0.22	0.23	0.21	0.17	0.26	0.20	0.15	0.19	0.16
C21:0	0.07	0.06	0.06	0.09	0.04	0.08	0.07	0.09	0.07	0.07	0.07	0.07	0.05	0.08	0.07
C22:0	0.15	0.14	0.11	0.11	0.07	0.19	0.13	0.14	0.13	0.11	0.16	0.12	0.10	0.12	0.10
C23:0	0.04	0.04	0.04	0.04	0.02	0.06	0.03	0.03	0.03	0.02	0.05	0.04	0.03	0.03	0.03
C24: 0	0.09	0.09	0.08	0.07	0.05	0.11	0.08	0.08	0.07	0.06	0.1	0.09	0.07	0.07	0.05
Total SFAs	9.96	13.1	19.5	27.7	27.7	11.4	16.1	29.5	35.4	41.1	15.7	37.0	25.6	56.8	65.3
*Monounsaturated fatty acids (MUFAs)*															
C11:1ω-1	1.47	1.53	1.43	0.67	1.12	1.36	1.33	1.28	1.25	1.03	1.20	1.32	1.13	1.17	0.96
C12:1ω-7	0.01	0.01	0.02	0.02	0.02	0.01	0.02	0.02	0.03	0.03	0.01	0.02	0.02	0.03	0.04
C14:1ω-5	0.01	0.02	0.03	0.04	0.04	0.02	0.02	0.04	0.06	0.06	0.01	0.03	0.04	0.07	0.08
C16:1ω-7t	0.00	0.01	0.01	0.02	0.02	0.01	0.01	0.02	0.03	0.02	0.01	0.01	0.01	0.02	0.03
C16:1ω-7	0.24	0.32	0.46	0.65	0.59	0.32	0.36	0.66	0.90	0.99	0.37	0.60	0.58	1.07	1.17
C17:1ω-7	0.02	0.02	0.02	0.02	0.01	0.02	0.02	0.02	0.02	0.02	0.02	0.02	0.02	0.02	0.02
C18:1ω-9t	0.03	0.04	0.04	0.08	0.04	0.04	0.04	0.03	0.03	0.04	0.07	0.04	0.02	0.04	0.04
C18:1ω-7t	0.07	0.08	0.09	0.07	0.10	0.09	0.05	0.08	0.11	0.12	0.16	0.17	0.08	0.12	0.14
C18:1ω-9	24.5	21.7	17.1	14.5	10.1	33.4	20.2	20.6	17.5	12.6	22.3	14.3	11.8	14.1	10.9
C18:1ω-7	1.34	1.17	0.91	0.75	0.52	1.86	1.18	1.16	1.01	0.72	1.18	0.77	0.62	0.73	0.56
C20:1ω-15	0.10	0.09	0.11	0.04	0.09	0.12	0.11	0.10	0.09	0.09	0.09	0.10	0.05	0.09	0.06
C20:1ω-12	0.03	0.02	0.02	0.02	0.02	0.02	0.03	0.02	0.02	0.01	0.02	0.02	0.02	0.02	0.02
C20:1ω-9	0.45	0.40	0.32	0.26	0.18	0.61	0.36	0.37	0.32	0.23	0.35	0.23	0.20	0.23	0.18
C22:1ω-9	0.02	0.02	0.02	0.02	0.01	0.03	0.01	0.02	0.02	0.02	0.02	0.01	0.01	0.01	0.01
C24:1ω-9	0.08	0.07	0.06	0.06	0.04	0.10	0.07	0.07	0.07	0.06	0.06	0.05	0.05	0.05	0.05
Total MUFAs	28.4	25.5	20.6	17.2	12.9	38.0	23.8	24.5	21.4	16.0	25.9	17.7	14.6	17.8	14.2
*Polyunsaturated fatty acids (PUFAs)*															
C16:2ω-4	0.01	0.01	0.01	0.01	0.01	0.01	0.01	0.00	0.01	0.01	0.01	0.01	0.01	0.01	0.01
C16:3ω-4	0.00	0.00	0.02	0.02	0.01	0.00	0.03	0.03	0.01	0.02	0.00	0.01	0.01	0.02	0.01
C18:2ω-6t	0.01	0.01	0.01	0.01	0.01	0.01	0.01	0.01	0.01	0.01	0.02	0.02	0.01	0.01	0.01
C18:2ω-6	16.3	16.3	15.1	11.2	13.6	18.1	15.0	16.3	17.5	15.2	20.5	20.2	17.6	19.5	16.0
C18:3ω-6	0.02	0.01	0.01	0.01	0.00	0.00	0.00	0.01	0.01	0.01	0.01	0.01	0.01	0.01	0.01
C18:3ω-4	0.14	0.11	0.07	0.05	0.02	0.20	0.11	0.10	0.07	0.04	0.10	0.05	0.04	0.04	0.02
C18:3ω-3	3.90	3.43	2.71	2.20	1.56	5.34	3.32	3.41	2.87	2.02	3.03	1.98	1.80	2.07	1.69
C18:4ω-1	0.01	0.01	0.01	0.01	0.01	0.02	0.01	0.01	0.01	0.01	0.01	0.01	0.01	0.01	0.01
C20:2ω-6	0.04	0.04	0.04	0.04	0.03	0.06	0.04	0.04	0.04	0.03	0.04	0.04	0.03	0.04	0.03
C20:3ω-9	0.01	0.01	0.01	0.01	0.02	0.03	0.00	0.01	0.00	0.00	0.02	0.03	0.03	0.02	0.00
C20:3ω-6	0.01	0.01	0.01	0.01	0.01	0.01	0.01	0.01	0.01	0.01	0.01	0.01	0.01	0.01	0.02
C20:4ω-6	0.03	0.03	0.03	0.04	0.03	0.04	0.02	0.02	0.03	0.03	0.03	0.02	0.02	0.03	0.03
C20:3 ω-3	0.02	0.02	0.02	0.01	0.01	0.02	0.01	0.01	0.01	0.01	0.01	0.02	0.01	0.01	0.01
C20:5 ω-3	0.01	0.01	0.02	0.02	0.02	0.03	0.01	0.02	0.02	0.02	0.01	0.02	0.02	0.03	0.04
C22:2 ω-6	0.01	0.01	0.01	0.01	0.00	0.04	0.01	0.01	0.00	0.00	0.01	0.01	0.01	0.01	0.02
C22:5 ω-3	0.01	0.01	0.01	0.01	0.01	0.01	0.00	0.00	0.01	0.01	0.01	0.01	0.00	0.01	0.01
C22:6 ω-3	0.02	0.02	0.02	0.02	0.01	0.02	0.02	0.02	0.02	0.01	0.02	0.02	0.02	0.02	0.02
Total PUFAs	20.6	20.0	18.1	13.7	15.4	23.9	18.5	20.0	20.6	17.5	23.9	22.4	19.6	21.9	17.9
Ʃ PUFA ω-3	3.95	3.49	2.77	2.27	1.61	5.42	3.36	3.46	2.93	2.08	3.09	2.05	1.84	2.14	1.76
Ʃ PUFA ω-6	16.4	16.4	15.2	11.3	13.7	18.3	15.0	16.3	17.6	15.3	20.7	20.3	17.6	19.6	16.1
ω-6/ω-3	4.16	4.69	5.49	4.98	8.47	3.37	4.47	4.72	6.00	7.37	10.7	12.2	11.5	10.6	9.50

^1^ The starter diets were provided to broilers from day 2 to day 10 of the experiment and the treatment groups were five increasing levels of Black Soldier Fly larvae (BSFL) dietary inclusions: 0, 2.5, 5, 7.5, and 10%; ^2^ the grower diets were provided to broilers from day 11 to day 21 of the experiment and the five levels of BSFL dietary inclusions were 0, 5, 10, 15, and 20%; ^3^ the finisher diets were provided to broilers from day 22 to day 42 of the experiment and the five levels of BSFL dietary inclusions were 0, 5, 10, 15, and 20%.

**Table 2 foods-10-00297-t002:** Summary statistics of live animal, commercial, and computed tomography (CT) compositional carcass traits and cut yield for broilers fed diets including the maximum inclusion levels of 0%, 5%, 10%, 15%, and 20% of Black Soldier Fly larvae (BSFL).

Trait	T1		T2		T3		T4		T5	
	Mean ± S.E.	Range	Mean ± S.E.	Range	Mean ± S.E.	Range	Mean ± S.E.	Range	Mean ± S.E.	Range
BW ^1^, *g*	3084.98 ± 60.85	2321–3448	3034.45 ± 63.56	2333–3503	3099.80 ± 63.56	2248–3600	3201.65 ± 68.39	2534–3724	3282.34 ± 66.66	2842–3958
	CT scanned compositional carcass traits									
	N = 23		N = 23		N = 21		N = 20		N = 21	
Muscle, g	1613.8 ± 35.9	1203–1884	1603.9 ± 34.6	1181–1813	1667.9 ± 32.8	1242–1878	1709.4 ± 39.5	1275–2012	1741.5 ± 39.6	1433–2116
Total fat, g	901.3 ± 20.0	675–1062	879.9 ± 20.0	675–1043	900.3 ± 17.3	737–1064	940.7 ± 23.5	658–1133	949.9 ± 19.0	811–1138
Total bone, g	152.2 ± 3.6	124–185	154.4 ± 4.06	109–180	167.9 ± 4.59	119–202	166.5 ± 5.05	134–218	168.8 ± 4.13	131–203
Muscle/BW	0.53 ± 0.01	0.44–0.58	0.53 ± 0.00	0.51–0.55	0.54 ± 0.00	0.50–0.57	0.54 ± 0.00	0.50–0.59	0.53 ± 0.01	0.44–0.59
Fat/BW	0.29 ± 0.00	0.5–0.58	0.29 ± 0.00	0.51–0.55	0.29 ± 0.00	0.52–0.57	0.29 ± 0.00	0.5–0.57	0.29 ± 0.01	0.44–0.61
Cut yield traits										
	N = 24		N = 24		N = 24		N = 24		N = 24	
Breast, g	327.8 ± 11.3	180–407.2	320.7 ± 7.69	234–387	315.9 ± 11.1	185–408	331.0 ± 9.78	265–399	342.1 ± 10.3	253–454
Drumstick, g	259.2 ± 4.56	216 ± 300	249.2 ± 4.35	216–298	248.6 ± 5.98	196–305	271.7 ± 7.89	189–341	276.8 ± 4.82	235–334
Thigh, g	301.4 ± 8.45	195 ± 362	297.6 ± 7.52	247–366	291.2 ± 7.81	201–382	319.2 ± 7.81	245–398	319.1 ± 7.81	271–404
Breast/BW	0.11 ± 0.00	0.08–0.13	0.11 ± 0.00	0.09–0.15	0.10 ± 0.00	0.07–0.16	0.10 ± 0.00	0.09–0.13	0.10 ± 0.00	0.08–0.14
Drumstick/BW	0.08 ± 0.00	0.07–0.10	0.08 ± 0.00	0.07–0.10	0.08 ± 0.00	0.07–0.10	0.08 ± 0.00	0.07–0.10	0.09 ± 0.00	0.08–0.12
Thigh/BW	0.10 ± 0.00	0.08–0.13	0.10 ± 0.00	0.08–0.12	0.10 ± 0.00	0.08–0.12	0.10 ± 0.00	0.08–0.11	0.10 ± 0.00	0.09–0.11

^1^ BW = the final body weight (in grams) of broilers at day 42. S.E. = standard error; T1 = the control group, no inclusion of BSFL in the diet; T2 = the 2.5% BSFL starter period followed by 5% BSFL grower and finisher periods; T3 = the 5% BSFL starter period followed by 10% BSFL grower and finisher periods; T4 = the 7.5% BSFL starter period followed by 15% BSFL grower and finisher periods; T5 = the 10% BSFL started period followed by 20% BSFL grower and finisher periods. Appendix A also represent these data.

**Table 3 foods-10-00297-t003:** Predicted least square means of carcass composition of broiler chickens at 42 days of age fed different levels of Black Soldier Fly larvae (BSFL).

Parameter	T1	T2	T3	T4	T5	S.E.M.
Muscle, g	1649	1654	1676	1680	1666	23.4
Total fat, g	920	907	905	925	910	13.6
Total bone, g	155 ^a^	159	169 ^b^	164	162	4.70
Muscle/BW ^1^	0.28	0.28	0.29	0.29	0.28	0.01
Fat/BW	0.09	0.08	0.08	0.09	0.08	0.02

^1^ BW = the final body weight (in grams) of broilers at day 42. S.E.M. = standard error of the mean; T1 = the control group, no inclusion of BSFL in the diet; T2 = the 2.5% BSFL starter period followed by 5% BSFL grower and finisher periods; T3 = the 5% BSFL starter period followed by 10% BSFL grower and finisher periods; T4 = the 7.5% BSFL starter period followed by 15% BSFL grower and finisher periods; T5 = the 10% BSFL started period followed by 20% BSFL grower and finisher periods; ^a,b^ within rows, means with a different superscript differ at (*p* < 0.05).

**Table 4 foods-10-00297-t004:** Predicted least square means of cut yield of broiler chickens at 42 days of age fed different levels of Black Soldier Fly larvae (BSFL).

Parameter	T1	T2	T3	T4	T5	S.E.M.
Breast, g	332	330	323	325	328	11.8
Drumstick, g	261	255	253	268	268	6.25
Thigh, g	305	306	298	313	306	8.20
Breast/BW ^1^	0.01	0.01	0.01	0.01	0.01	0.0008
Drumstick/BW	0.01	0.01	0.01	0.01	0.01	0.0004
Thigh/BW	0.01	0.01	0.01	0.01	0.01	0.001

^1^ BW = the final body weight (in grams) of broilers at day 42. T1 = the control group, no inclusion of BSFL in the diet; T2 = the 2.5% BSFL starter period followed by 5% BSFL grower and finisher periods; T3 = the 5% BSFL starter period followed by 10% BSFL grower and finisher periods; T4 = the 7.5% BSFL starter period followed by 15% BSFL grower and finisher periods; T5 = the 10% BSFL started period followed by 20% BSFL grower and finisher periods.

**Table 5 foods-10-00297-t005:** Predicted least square means of breast meat quality parameters of broiler chickens at 42 days of age fed different levels of Black Soldier Fly larvae (BSFL).

Parameters	T1	T2	T3	T4	T5	S.E.M.
pH	5.74	5.78	5.76	5.75	5.71	0.31
*L**	58.2	59.4	58.7	59.0	59.4	0.78
*a**	4.63	4.52	4.45	4.51	4.21	0.29
*b**	0.42	0.67	0.75	0.72	0.90	0.25
*C**	4.71	4.67	4.57	4.69	4.38	0.27
*h**	0.09	0.15	0.17	0.16	0.22	0.06
Lipid oxidation, mg MDA/kg ^1^	0.72	0.75	0.72	0.71	0.71	0.03
Cooking loss, %	31.4	30.3	30.2	29.8	29.9	0.97
Shear force, N	23.4	22.4	20.1	22.1	21.1	2.26

T1 = the control group, no inclusion of BSFL in the diet; T2 = the 2.5% BSFL starter period followed by 5% BSFL grower and finisher periods; T3 = the 5% BSFL starter period followed by 10% BSFL grower and finisher periods; T4 = the 7.5% BSFL starter period followed by 15% BSFL grower and finisher periods; T5 = the 10% BSFL started period followed by 20% BSFL grower and finisher periods; ^1^ malondialdehyde per kg of wet sample.

**Table 6 foods-10-00297-t006:** Predicted least square means of the amino acid profile of breast muscle obtained from broiler chickens at 42 days of age fed different levels of Black Soldier Fly larvae (BSFL).

Amino Acid, mg/g	T1	T2	T3	T4	T5	S.E.M.
Histidine	24.6	23.2	23.6	24.2	24.4	0.52
Serine	34.3 ^a^	33.5 ^ab^	32.9 ^b^	33.3 ^ab^	33.8 ^ab^	0.46
Arginine	55.0	53.5	53.5	53.7	53.8	0.77
Glycine	37.7	37.3	37.1	37.5	37.3	0.49
Aspartic acid	67.5 ^b^	73.3 ^a^	72.6 ^a^	73.1 ^a^	71.1 ^ab^	1.39
Glutamine	114 ^b^	121 ^a^	121 ^a^	121 ^a^	119 ^ab^	1.87
Threonine	38.6	37.5	37.3	37.7	38.0	0.55
Alanine	43.7	45.1	45.0	45.2	44.8	0.63
Proline	29.8	30.0	29.8	30.2	30.2	0.38
Lysine	68.6 ^a^	72.1 ^b^	72.2 ^b^	72.5 ^b^	70.9 ^ab^	1.16
Tyrosine	28.5	27.7	27.6	27.8	28.1	0.41
Methionine	24.5	23.8	23.9	24.0	24.0	0.36
Valine	42.0	41.5	41.9	42.0	41.6	0.63
Isoleucine	41.0	40.5	41.0	40.9	40.5	0.60
Leucine	68.2	67.3	67.2	67.5	67.5	0.95
Phenylalanine	34.5	33.6	33.4	33.6	33.6	0.48
Total	752	761	760	764	758	10.2

T1 = the control group, no inclusion of BSFL in the diet; T2 = the 2.5% BSFL starter period followed by 5% BSFL grower and finisher periods; T3 = the 5% BSFL starter period followed by 10% BSFL grower and finisher periods; T4 = the 7.5% BSFL starter period followed by 15% BSFL grower and finisher periods; T5 = the 10% BSFL started period followed by 20% BSFL grower and finisher periods; ^a,b^ within rows, means with a different superscript differ at (*p* < 0.05).

**Table 7 foods-10-00297-t007:** Predicted least square means of fatty acids composition of breast muscle obtained from broiler chickens at 42 days of age fed different levels of Black Soldier Fly larvae (BSFL).

Fatty Acid, mg/100 g	T1	T2	T3	T4	T5	S.E.M.
*Saturated fatty acids (SFAs)*						
C8:0	5.85	6.29	6.39	6.54	6.92	0.72
C10:0	5.78 ^b^	7.29 ^b^	7.89 ^ab^	10.0 ^a^	9.91 ^a^	0.82
C11:0	1.36	1.11	1.13	1.27	1.36	0.13
C12:0	12.2 ^c^	81.4 ^cb^	139.9 ^b^	221.4 ^a^	268.8 ^a^	27.5
C14:0	18.7 ^c^	42.9 ^cb^	62.3 ^b^	88.0 ^a^	103.0 ^a^	9.88
C15:0anteiso	0.10 ^c^	0.20 ^cb^	0.30 ^b^	0.50 ^a^	0.61 ^a^	0.05
C15:0	2.03	2.35	2.47	2.76	2.75	0.29
C16:0	475	512	459	454	418	56.3
C17:0iso	0.39 ^c^	0.55 ^cb^	0.89 ^b^	1.35 ^a^	1.75 ^a^	0.15
C17:0	3.95	4.24	3.91	3.95	3.96	0.54
C18:0	213 ^a^	215 ^a^	173 ^ab^	161 ^b^	146 ^b^	17.7
C20:0	2.76 ^a^	2.68 ^a^	2.16 ^ab^	2.05 ^b^	1.92 ^b^	0.19
C21:0	1.47	1.66	1.69	1.72	1.73	0.20
C22:0	1.44 ^a^	1.36 ^ab^	1.24 ^b^	1.21 ^b^	1.21 ^b^	0.06
C23:0	0.36	0.33	0.36	0.32	0.32	0.03
C24:0	0.93	0.85	0.80	0.79	0.81	0.05
Ʃ SFA	736 ^b^	789 ^ab^	833 ^ab^	871 ^ab^	970 ^a^	79.4
*Monounsaturated fatty acids (MUFAs)*						
C11:1ω-1	0.50	0.55	0.42	0.42	0.61	0.10
C14:1ω-5	1.41 ^c^	2.86 ^cb^	5.51 ^ba^	8.61 ^a^	11.5 ^a^	1.08
C16:1ω-7	30.7 ^b^	43.6 ^b^	54.2 ^ab^	65.7 ^a^	70.4 ^a^	8.16
C16:1ω-7t	0.29 ^b^	0.36 ^b^	0.41 ^b^	0.59 ^a^	0.59 ^a^	0.08
C17:1ω-7	6.19	5.34	5.42	5.64	4.94	0.43
C18:1ω-7t	2.93 ^a^	2.36 ^ab^	1.76 ^ab^	1.64 ^b^	1.61 ^b^	0.44
C18:1ω-9	433	452	422	415	383	52.7
C18:1ω-9t	4.40 ^a^	5.06 ^a^	3.08 ^b^	3.14 ^b^	3.44 ^b^	0.39
C19:1ω-12	0.14	0.12	0.13	0.11	0.11	0.01
C20:1ω-9	5.42	5.74	5.18	5.20	5.10	0.58
C20:1ω-12	0.63	0.71	0.51	0.68	0.67	0.10
C20:1ω-15	0.27	0.31	0.35	0.34	0.32	0.04
C22:1ω-9	0.45	0.46	0.47	0.49	0.49	0.03
C24:1ω-9	1.64	1.64	1.73	1.88	1.88	0.09
Ʃ MUFA	517	551	535	545	521	64.5
*Polyunsaturated fatty acids (PUFAs)*						
C16:2ω-4	0.45 ^a^	0.42 ^a^	0.42 ^a^	0.30 ^ab^	0.21 ^b^	0.05
C16:3ω-4	0.47	0.31	0.25	0.29	0.28	0.08
C18:2ω-6	565 ^a^	558 ^a^	488 ^a^	416 a^b^	326 ^b^	56.3
C18:3ω-3	19.1	19.8	18.6	19.3	17.8	2.66
C18:3ω-4	0.50 ^a^	0.37 ^a^	0.37 ^a^	0.29 ^b^	0.19 ^b^	0.06
C18:3ω-6	4.11	4.03	3.60	3.28	2.94	0.54
C18:4ω-3	0.73	0.72	0.89	0.77	0.76	0.10
C20:2ω-6	13.5 ^a^	12.9 ^a^	11.1 ^ab^	9.95 ^bc^	8.17 ^c^	0.66
C20:3ω-6	10.6	10.9	10.5	10.5	10.6	0.58
C20:3ω-9	1.34 ^c^	1.55 ^c^	1.50 ^c^	1.71 ^b^	2.11 ^a^	0.12
C20:4ω-6	59.2 ^a^	55.0 ^a^	56.9 ^a^	51.9 ^b^	45.6 ^b^	2.48
C20:5ω-3	1.38 ^b^	1.54 ^ab^	1.62 ^ab^	1.90 ^a^	2.51 ^ab^	0.12
C22:2ω-6	0.49 ^a^	0.44	0.45	0.40	0.33 ^b^	0.04
C22:4ω-6	16.6 ^a^	14.9 ^a^	15.2 ^a^	12.8 ^b^	10.9 ^b^	0.76
C22:5ω-3	8.80	8.37	8.48	8.38	8.40	0.45
C22:5ω-6	4.47 ^a^	4.02 ^a^	3.93 ^a^	3.07 ^b^	2.44 ^b^	0.36
C22:6ω-3	6.49	5.55	5.66	5.63	5.71	0.46
Ʃ PUFA ω*-3*	37.8	37.4	36.6	37.4	36.7	2.98
Ʃ PUFA ω*-6*	674 ^a^	660 ^a^	590 ^a^	508 ^ab^	407 ^b^	58.7
ω*-6*/ω*-3*	17.6 ^a^	17.6 ^a^	15.8 ^b^	13.5 ^c^	11.2 ^d^	0.54
Ʃ PUFA	715 ^a^	701 ^a^	630 ^a^	550 ^ab^	448 ^b^	61.8
	*Trans-fatty acids*					
Σ TFA	8.27 ^a^	8.35 ^a^	5.77 ^b^	5.80 ^ab^	5.83 ^ab^	0.90
			*Indexes*			
Atherogenic index	0.46 ^e^	0.60 ^d^	0.73 ^c^	0.92 ^b^	1.13 ^a^	0.03
Thrombotic index	0.03 ^e^	0.06 ^d^	0.09 ^c^	0.13 ^b^	0.17 ^a^	0.01

T1 = the control group, no inclusion of BSFL in the diet; T2 = the 2.5% BSFL starter period followed by 5% BSFL grower and finisher periods; T3 = the 5% BSFL starter period followed by 10% BSFL grower and finisher periods; T4 = the 7.5% BSFL starter period followed by 15% BSFL grower and finisher periods; T5 = the 10% BSFL started period followed by 20% BSFL grower and finisher periods; ^a,b,c,d,e^ within rows, means with a different superscript differ at (*p* < 0.05).

## Data Availability

The remaining data are available on request from the corresponding author.

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
