# Peer review of "Technological Quality, Amino Acid and Fatty Acid Profile of Broiler Meat Enhanced by Dietary Inclusion of Black Soldier Fly Larvae"

_foods, 2021, doi:10.3390/foods10020297_

Round 1

Reviewer 1 Report

The article Foods 1054486 “Fatty acid composition of broiler breast meat enhanced 3 by dietary inclusion of Black Soldier Fly larvae” is an interesting article evaluating the influence of black soldier larvae's inclusion the chickens feed (to a maximum of 20%). Several quality parameters were evaluated. A particular attention was given to the fatty acid composition, which was influenced by the feeding with BSFL.

The experiment was well organized, with a robust number of samples, and an analytical plan very well established.

The results are presented in an organized form and discussed in a pragmatic way, which is very positive for understanding the results.

The discussion around the fatty acids is more complex, and authors argue to potential benefits of the chicken fatty acids for consumer health. This approach is very honest and correctly focused on the facts.

The authors demonstrated the possibility of using BFSL in feeding chicken without the risk of developing abnormal quality characteristics, contributing to a growing trend of using insects to animal feeding taking advantage of the economic and environmental benefits of that feed.

I found the article very well written, straightforward, and easy to understand.

Only two small notes:

Line 284. myristelaic – Myristoleic?

Line 324-325- Consider to use the unit symbol instead of the descriptive for Newtons

Reviewer 2 Report

Manuscript revision ID: foods-71054486 titled:

Fatty acid composition of broiler breast meat enhanced by dietary inclusion of Black Soldier Fly larvae

The content of the manuscript is more suitable for Animals than for Foods. Describes the results of a poultry feeding experiment resulting in the quality and chemical composition of meat, but more in terms of the BSFL nutritional factor. In the title, the authors mention FA, and the research also concerns slaughter efficiency, AA composition and technological quality of meat - this should also be included in the title.

Introduction

L40-41- Please describe the difference in the chemical composition and fatty acids of soybeans and BSFL - if the authors mention its replacement in poultry nutrition.

L59-60 – The fatty acid composition of the animal diets also significantly models the quality of fat in the animal's meat tissues.

M@M

L85 – Please provide the chemical composition (amino acid and fatty acid composition) of the feed additive - BSFL. Please also provide the quantitative component composition (which feeds) of the experimental mixtures, their nutritional value and chemical composition as far as analyzed in the muscles. What was the protein and energy level? Were the mixtures iso-protein and isoenergetic? To what extent did they capture the bird's needs? - this is the basic information for describing a feeding experience.

Table 1. – this is not: "Nutrient composition and fatty acid profile of experimental diets ...", but only - Fatty acid profile - please correct it.

Table 2 - I propose to break it down into two parts: CT scanned compositional carcass traits and Cut yield traits. The analysis of predicted least square means from table 3 and 4 should be attached to them. This will allow avoiding duplication of results to some extent.

Table 5-7 – I recommend basing on the measurement averages and possibly supplementing them with the analysis predicted least square means….

Please complete the tables with the results of the statistical analysis – P-value.

Table 6. – Sum of SFA - the difference between T1 and T5 is almost 30% - and it was not statistically significant ? SEM - 105? quite high ….

Disscusion

L315-320 – please refer to the research similar to that carried out by the Authors in the discussion. What was the amino acid composition of BFSL? It is difficult to ponder about the possibility of synthesizing individual AAs if we do not know the size of their supply in feed….

L362-377 – a very bold statement ... However, the authors do not emphasize that BSFL increases the SFA pool by about 30% and this is unfavorable for consumers.

Conclusion

L458 – I would consider such an unequivocal conclusion. The 20% BSFL content in the mix significantly increased the SFA content of meat that people eat too much and it is recommended to limit its level in the diet. Will it balance the increased level of lauric acid in health terms? I would be careful. For production, I'd rather recommend a slightly lower BSFL level, and I would test as high as 20% of BSFL level in diets further….

Reviewer 3 Report

Although it has not been studied, the main effect of including a new ingredient in animal feed for consumption should be a sensory analysis of consumers. The modification of the lipid profile of a meat can generate changes in the taste of the consumers that deserve to be studied.

If the main effects of including BSFL in the broilers diet are anti-inflammatory and other positive health effects, as has been refered in the introduction by the authors, why have they not been studied?

The use of the TC methodology to estimate the composition of broiler carcasses is an expensive method, mainly because it has been known since the last century that the diet has no effect on carcass quality, resulting in a small impact in the present study. The estimation errors show the inferior effectiveness of the method in relation to the traditional dissection, which in chickens is not as expensive as in other species in which methodologies based on non-destructive images and spectroscopic techniques are justified. In reality, the use of the tomographic technique is not very suitable for the purpose of this study on the fatty acid composition of broiler breast meat enhanced by dietary inclusion of BSFL. 

The color attributes (C* and h*) should be calculated and discussed. We find it strange that in relation to the yellowness index that with such a wide range of variation there were no differences between treatments recorded.

In terms of lipid quality the index of atherogenicity (IA) and the index of thrombogenicity (IT) as well as the ∑ trans fatty acids would be interesting to present and discuss.

The results of TBARS should be discussed mainly given the increase in the amount of saturated fat in chickens fed with BSFL. It would have been important to study the effect of BSFL on the meat quality due to the storage time.

Round 2

Reviewer 3 Report

Results of trans-saturated fatty acids, atherognicity, and thrombogenicity indexes should be discussed.
